# The Female Reproductive Tract Microbiota and Endometrial Cancer: A Systematic Review

**DOI:** 10.3390/ijms26188877

**Published:** 2025-09-12

**Authors:** Riccardo Vizza, Francesco Belli, Paolo Fabene, Valentina Salari, Chiara Casprini, Giacomo Corrado, Antonio Simone Laganà, Pier Carlo Zorzato, Mariachiara Bosco, Irene Porcari, Stefano Uccella, Simone Garzon

**Affiliations:** 1Unit of Obstetrics and Gynecology, Department of Surgery, Dentistry, Pediatrics, and Gynecology, AOUI Verona, University of Verona, 37125 Verona, Italy; riccardo.vizza@studenti.univr.it (R.V.); francesco.belli_02@studenti.univr.it (F.B.); chiara.casprini@studenti.univr.it (C.C.); piercarlo.zorzato@univr.it (P.C.Z.); mariachiara.bosco@univr.it (M.B.); irene.porcari@aovr.veneto.it (I.P.); stefano.uccella@univr.it (S.U.); 2Department of Engineering, Innovation Medicine Faculty of Medicine, University of Verona, 37125 Verona, Italy; paolo.fabene@univr.it (P.F.); valentina.salari@univr.it (V.S.); 3Department of Excellence in Neurosciences, Biomedicine, and Movement Science, School of Medicine, University of Verona, 37124 Verona, Italy; 4UOC Ginecologia Oncologica, Dipartimento di Scienze per la Salute della Donna e del Bambino e di Sanità Pubblica, Fondazione Policlinico Universitario A. Gemelli, IRCCS, 00136 Rome, Italy; giacomo.corrado@policlinicogemelli.it; 5Unit of Obstetrics and Gynecology, “Paolo Giaccone” Hospital, Department of Health Promotion, Mother and Child Care, Internal Medicine and Medical Specialties (PROMISE), University of Palermo, 90127 Palermo, Italy; antoniosimone.lagana@unipa.it

**Keywords:** endometrial cancer, microbiota, microbial diversity

## Abstract

This systematic review aimed to summarize the available evidence on the associations between the female reproductive tract microbiota and endometrial cancer (EC). While gut microbiota has been studied extensively, microbial communities within the endometrium, cervix, and vagina remain relatively understudied. A systematic literature search was conducted in PubMed, Scopus, Web of Science, EMBASE, and the Cochrane Library for studies published up to January 2025. Predefined PECO-based criteria included studies on women or human cell models assessing genital tract microbiota in EC versus non-EC controls, focusing on composition, diversity, or function; reviews and non-microbiota studies were excluded. A total of 21 studies were included. Overall, there was a consistent depletion of protective *Lactobacillus* species and enrichment of anaerobic, pro-inflammatory bacteria like *Prevotella*, *Atopobium*, and *Porphyromonas* in EC tissues. Beta-diversity was significantly different between EC and controls across studies, indicating distinct microbial profiles. Some studies also identified fungal and viral taxa associated with EC, and functional assays demonstrated that certain species could modulate host immune responses or promote tumor growth. Despite methodological heterogeneity and the lack of evidence supporting causality due to the observational design, the findings support an association between altered genital tract microbiota and EC.

## 1. Introduction

The microbiota is the collection of all microbes, including bacteria, archaea, fungi and viruses, that naturally live in and on our bodies. Technically, the term microbiota refers to the community of microorganisms themselves, whereas the microbiome refers to the collective genomes of these microorganisms. These communities colonize various anatomic regions, such as the gastrointestinal, oral, and genitourinary tracts, and can be classified as symbionts, commensals, or pathogens. Microbial communities vary depending on the anatomical site and are influenced by factors including health status, genetics, diet, and hygiene [1].

Several studies suggest that the microbiota is essential in the immune and endocrine systems [2,3]. Furthermore, microbiota alterations have been associated with the development of diseases, including cancer [4]. Dysbiosis, or imbalances in the intestinal and vaginal microbiota, appears to be associated with the onset of gynecological malignancies, including uterine, cervical, vulvar, and vaginal cancers [5]. Specifically, recent evidence suggests that the gut microbiota plays a significant role in the development and progression of endometrial cancer (EC) through several mechanisms, including immune modulation, metabolite production, and alteration of the inflammatory environment [6].

EC is the most common gynecological cancer in high-income countries and is the fourth most common cancer among women, following breast, lung, and colorectal cancers [7]. Major risk factors include genetic predisposition, hormonal imbalances involving hypoestrogenism, prolonged exposure to estrogen, obesity, nulliparity, and advanced age. Variations in the gut microbiota have been closely linked to the onset of EC [8]. Notably, diet is one of the primary factors influencing the microbiota composition. Diets rich in fiber and probiotics appear to exert a protective effect by reducing systemic inflammation, enhancing immune function, and maintaining normal estrogen levels [9]. In contrast, a high-fat, high-calorie diet may cause dysbiosis, promote systemic inflammation, and alter estrogen metabolism. This concept aligns with the emerging notion of the estrobolome, the collection of gut microbial genes capable of metabolizing estrogens, whereby increased microbial β-glucuronidase activity may enhance enterohepatic recirculation of estrogens and, in turn, hypothetically contribute to endometrial carcinogenesis [10].

Based on this evidence, the gut microbiota may play a significant role in initiating and progressing EC through multiple mechanisms, including immune modulation, metabolite production, and inflammation regulation [9,10]. However, despite growing attention to the intestinal microbiota, other microbial ecosystems, particularly those anatomically and patho-physiologically closer to the endometrium, have received considerably less focus. For a long time, the uterus was believed to be a sterile environment. However, advancements in detection technologies have since revealed the presence of a resident microbial community within the endometrium [11]. In this context, the lower and upper genital tract microbiota, directly adjacent to the endometrial environment, represent promising but understudied areas in EC pathogenesis. Therefore, this systematic review aims to summarize the available evidence on the female reproductive tract microbiota and its association with the pathogenesis of EC.

## 2. Materials and Methods

### 2.1. Study Design

This systematic review was designed before the online search, outlining the study population, exposure, comparator, outcome measures, and study eligibility criteria. The study was exempt from institutional review board approval. The methodology followed the Cochrane Handbook for Systematic Reviews of Diagnostic Test Accuracy. The review was reported following the Preferred Reporting Items for Systematic Reviews and Meta-Analyses of Diagnostic Test Accuracy (PRISMA-DTA) guidelines [12,13]. The protocol was registered on PROSPERO (CRD 420251024643).

### 2.2. Search Strategy and Eligibility Criteria

A comprehensive literature search was conducted across EMBASE, Scopus, PubMed, Web of Science, and the Cochrane Library from inception until January 2025 by a certified professional librarian (Biblioteca Meneghetti—University of Verona). The search strategy was performed with and without MeSH terms to include non-indexed more recent records and to make the search query replicable in all electronic databases: (Endometrial Neoplasm OR Neoplasm, Endometrial OR Endometrial Carcinoma OR Carcinoma, Endometrial OR Cancer of Endometrium OR Endometrium Cancer OR Cancer, Endometrium OR Cancer of the Endometrium OR Carcinoma of Endometrium OR Endometrium Carcinoma OR Endometrial Cancer OR Cancer, Endometrial) AND (Microbiota OR Microbial Community OR Community, Microbial OR Microbial Community Composition OR Community Composition, Microbial OR Composition, Microbial Community OR Microbiome OR Human Microbiome OR Microbiome, Human OR Microbial Community Structure OR Community Structure, Microbial). Additionally, references from the selected studies were manually reviewed.

Inclusion and exclusion criteria were pre-defined by all contributors using the Population, Exposure, Comparison, and Outcome (PECO) framework. Population: Women or models based on human cells in vitro or ex vivo. Exposure: Diagnosis of EC or EC tissue, including all subtypes and stages and endometrial hyperplasia (EH). Comparison: Women or tissues without EC or EH, healthy or with non-neoplastic or pre-neoplastic pathologies (e.g., benign conditions). Outcome: Evaluation of the microbiota of the female genital tract (endometrium, cervix, vagina) conducted primarily through 16S rRNA gene sequencing, although other microbiological analysis methods were also considered. Any of the following microbiota evaluations were eligible: microbiota composition (taxonomy), microbial diversity (alpha/beta diversity), and functional activity of the microbiota (metabolic pathways, gene expression). Studies were excluded if they were review articles or did not include a microbiota analysis of the female genital tract. We harmonized diversity terminology across studies by defining α-diversity as within-sample richness/evenness (e.g., Shannon, Simpson, Chao1, Pielou’s evenness) and β-diversity as between-sample community dissimilarity (e.g., Bray–Curtis, Jaccard, weighted/unweighted UniFrac, Aitchison). For synthesis, we recorded the index/metric and classified results as ‘higher’, ‘lower’, or ‘no difference’ versus controls.

### 2.3. Study Selection and Data Extraction

Two independent reviewers (RV, FB) screened titles and abstracts for relevance and assessed the full texts of potentially eligible studies. Disagreements were resolved through discussions with a second reviewer (SG). A standardized data extraction form was developed to collect key study details, including study characteristics (first author, publication year, study design, research setting), sample data (population studied, sampling technique, sample size), and outcomes (taxonomy, alpha/beta diversity, metabolic pathways, gene expression, and clinical correlation). To minimize bias inherent to low-biomass microbiome studies, we prioritized reports using aseptic sampling (e.g., sterile, lubricant-free speculum with cervical cleansing) and laboratory contamination controls (negative controls/blank extractions), and we interpreted findings from studies that did not report these measures cautiously in the synthesis.

## 3. Results

A total of 597 records were initially retrieved through database searches. After removing 412 duplicates, 185 unique records were screened based on titles and abstracts. Of these, 129 studies were excluded due to irrelevance to the topic, non-original content, or lack of specific microbiota analysis. We then assessed 56 full-text articles for eligibility, excluding 35 that did not meet the inclusion criteria: 18 investigated the microbiota of anatomical sites other than the female genital tract (most commonly the intestinal microbiota), while the remaining 17 were excluded because they were not original articles.

Ultimately, 21 studies, published between 2016 and 2025, were included in the final synthesis (Figure 1). To facilitate a structured synthesis of evidence, we categorized them into three groups based on the type of analyzed biological sample. In the “Endometrial Microbiota in Endometrial Cancer” chapter, we included all studies that analyzed endometrial tissue samples. In “Cervicovaginal Microbiota in Endometrial Cancer” we included studies that analyzed vaginal or cervical tissue samples. In “Functional and in vitro studies,” we included studies that conducted in vitro experiments to assess microbial effects on host cells. Comprehensive summary table outlining the parameters used to identify the microbiota and JBI Analytical Cross-Sectional appraisal matrix are present in Appendix A.

### 3.1. Endometrial Microbiota in Endometrial Cancer

#### 3.1.1. Bacteria

We included 12 studies (Table 1) that compared the endometrial microbiota of patients with EC or endometrial hyperplasia to that of healthy individuals or patients with benign uterine pathologies [14,15,16,17,18,19,20,21,22,23,24]. Only one study stands out as an exception: Wang et al. compared the microbiota present in EC tissue with that found in the healthy peri-tumor tissue of the same patients [23].

These studies used α-diversity and β-diversity to assess the microbial diversity between groups: α-diversity refers to within-sample diversity, accounting for species richness and evenness. β-diversity quantifies the compositional dissimilarity between microbial communities from different patients.

All except two studies reported a significantly increased α-diversity in EC compared to controls, indicating a richer and more evenly distributed microbial community in malignant tissue [15,17,19,23,24]. In contrast, reduced α-diversity was observed in EC versus healthy controls in Li et al. [18] and in EH compared to benign controls in Ying [22], while Leoni et al. found no significant differences in α-diversity between EC and controls [20].

All studies but one reported significant differences in β-diversity between EC and controls, indicating distinct microbial community structures associated with malignancy [15,17,18,19,23,24]. Only Leoni et al. reported no significant group-level separation based on β-diversity metrics [20].

Regarding taxonomic signatures, EC tissues were consistently enriched in anaerobic and pro-inflammatory bacteria, with frequent detection of *Prevotella*, *Atopobium*, *Porphyromonas*, *Peptoniphilus*, and *Dialister* [15,18,19,23]. Han et al. and Wang et al. showed an increase in Fusobacterium and Sneathia [19,23], while Kuźmycz et al. specifically highlighted *Anaerococcus vaginalis* as a potential functional cofactor in EC, capable of inducing ROS in endometrial cells [24]. Streptococcus, Gardnerella, and Acinetobacter were found more often in EC than controls across multiple studies [17,18,20,23]. The most consistent finding across identified studies is the depletion of *Lactobacillus* spp., especially *L. crispatus* and *L. iners*, in EC compared to controls. These studies suggest that Lactobacilli are considered protective commensals, maintaining pH balance, epithelial barrier integrity, and suppressing pathogens [16,17,20,23]. Gonzalez-Bosquet et al. showed a reduction in EC of Desulfobacter, Desulfomicrobium, Parabacteroides, and Proteus compared to control samples [14].

Concerning EH, *Bacillus pseudofirmus*, *Stenotrophomonas*, and *Delftia* were identified as enriched genera with potential biomarker value [16,22].

A recent large-scale study by Liu et al. further expanded on these findings by analyzing the intratumoral microbiota of 239 EC patients and 116 controls using data from the TCGA-UCEC [25]. The authors identified 26 microbial taxa significantly associated with patient prognosis and used Cox and LASSO regression models to develop a microbiota-based risk score, suggesting a relationship between the prognosis of patients with EC and the composition of the intratumoral microbiota [25]. The low-risk group exhibited a microbial profile linked to higher immune activity and a better-predicted response to immune checkpoint inhibitors (PD1/PD-L1 and CTLA-4). In contrast, high-risk patients had elevated immune dysfunction and exclusion scores [25].

#### 3.1.2. Fungi and Viruses

Two studies investigated the role of fungi and viruses of endometrium in EC. Han et al. showed enrichment of *Penicillium* spp. in EC tissues, along with estradiol-like metabolites, suggesting fungal involvement in hormonal mimicry and tumor promotion [19]. Deligdisch et al. identified human mammary tumor virus (HMTV) sequences in 23% of EC samples but not in controls, proposing a possible viral contribution to endometrial carcinogenesis [26].

**Table 1 ijms-26-08877-t001:** Methodological characteristics and main results of studies on the endometrial microbiome.

First Author and Year	Sampling Method	Sample Size	Study Results
Deligdisch et al., 2013 [26]	Formalin-fixed paraffin-embedded (FFPE) tissues from hysterectomy and endometrial scraping; DNA sequencing performed.	EC n = 56, controls n = 33	HMTV env gene sequences and protein detected in 23.2% of EC cases, 0% in controls. Expression confirmed by nested PCR and immunohistochemistry, suggesting viral involvement in carcinogenesis.
Li et al., 2021 [18]	Endometrial tissue samples analyzed using 16S rRNA sequencing.	EC n = 30, controls n = 10	*Prevotella* and *Pelomonas* enriched in EC tissues. *Prevotella* abundance correlated with increased D-dimer and FDPs, and with genes involved in fibrin degradation (e.g., PRSS33, CPB2, XBP1). Combined markers (*Prevotella* + DD + FDPs) had high diagnostic potential (AUC = 0.86).
Chen et al., 2021 [15]	Endometrial biopsies analyzed using meta-transcriptomic sequencing.	EC n = 9 EC, normal group n = 8	Identified 5576 active bacterial species and 381 archaeal species in EC patients. Key species and pathways (e.g., Apelin, Wnt, IL-17) linked to tumor migration and host-microbiota metabolic crosstalk. Microbes potentially influence EMT and unfolded protein response. Among the most abundant species in the EC group: *Clostridium botulinum*, *Mycoplasma hyopneumoniae*, *Bacillus cereus*, *Pasteurella multocida*. 17 species showed significant differences between EC group and control group (e.g., Borrelia coriaceae ↑ in EC; Streptococcus mitis ↓ in EC.
Chao et al., 2022 [16]	Endometrial lavage fluid collected via transcervical catheter.	EH, n = 18; EC, n = 7; metastatic EC, n = 2; benign endometrial lesions, n = 8	Found over-representation of *Bacillus pseudofirmus* and *Stenotrophomonas rhizophila* in EC/EH patients. Suggested link between plastic-degrading bacteria and endometrial carcinogenesis. Microbiota function associated with fatty acid and amino acid metabolism
Kaakoush et al., 2022 [17]	Endometrial brushings or tissue biopsies analyzed via 16S rRNA sequencing.	EC n = 30, benign n = 30	Endometrial microbiota clustered into three community types. Cancer samples showed reduced *Lactobacillus*, with *Lactobacillus iners* enriched in controls. Obesity influenced community type prevalence but not *Lactobacillus* abundance. Similar microbiota between tumor and adjacent normal tissue.
Hawkins et al., 2022 [21]	Endometrial biopsies from patients undergoing hysterectomy, analyzed by 16S sequencing.	EC n = 30, benign n = 30	Higher microbial diversity in ECs from Black vs. White women. Tumors from Black women had more Firmicutes, Cyanobacteria, and OD1. *Lactobacillus* acidophilus enriched in Black women. Differences may contribute to racial disparities in EC outcomes.
Wang et al., 2022 [23]	Endometrial samples from hysterectomy procedures, 16S rRNA sequencing.	EC n = 28, pericancer n = 28	EC tissues showed higher alpha diversity and were enriched with *Prevotella*, *Atopobium*, *Anaerococcus*, *Dialister*, *Porphyromonas*, and *Peptoniphilus*. *Lactobacillus* dominated in adjacent non-EC tissues. Microbiota differences correlated with clinical stage, pH, and *Lactobacillus* abundance.
Leoni et al., 2024 [20]	Endometrial biopsies from patients undergoing hysterectomy, analyzed by 16S sequencing.	EC n = 8, controls n = 6	Confirmed low bacterial abundance in endometrium. Metabarcoding revealed higher prevalence of pathogenic genera in EC tissues. Cutibacterium more abundant in EC; Ralstonia more abundant in controls. No significant differences in diversity between groups.
Han et al., 2024 [19]	Endometrial tissue samples analyzed using 16S rRNA sequencing, and the ITS1 for the study of the uterine fungal microbiome.	EC n = 33, EH n = 15, benign n = 15.	EC and EH showed increased alpha diversity and shift in microbiome structure, especially fungal composition. Penicillium sp. enriched in EC/EH, Sarocladium in controls. Dysbiosis correlated with pro-inflammatory cytokines (IL-6, IL-11, TGF-β) and β-glucuronidase activity, implicating estrogen-like metabolic disruption.
Ying et al., 2024 [22]	Endometrial biopsies analyzed using 16S rRNA sequencing.	benign n = 53, EH n = 15 (including 2 AEH).	Patients with endometrial hyperplasia had significantly lower alpha diversity and increased abundance of *Delftia*, *Serratia*, and *Stenotrophomonas*. These bacteria showed diagnostic potential for EH with AUCs around 71–75%. Suggests potential for microbiota-based biomarkers.
Gonzalez-Bosquet et al., 2023 [14]	Tumor tissue samples analyzed via 16S rRNA.	EC n = 62, Controls n = 36, HSOC n = 112	Microbial diversity correlated with somatic variation. Specific bacterial taxa (e.g., Leclercia, Desulfobulbaceae) associated with high-grade serous ovarian cancer (HGSC) and endometrioid endometrial cancer (EEC). Pathway analyses suggested potential for early cancer detection.
Kuźmycz et al., 2025 [24]	Endometrial canal swabs collected pre-hysterectomy; 16S rRNA sequencing.	EC n = 16, endometrial myoma n = 13.	Higher microbial alpha- and beta-diversity in EC samples. *Anaerococcus* significantly enriched in EC and capable of adhering to uterine fibroblasts and inducing ROS production. Suggests a potential role in inflammation-mediated carcinogenesis.

### 3.2. Cervicovaginal Microbiota in Endometrial Cancer

We included six studies published between 2019 and 2024 that investigated the cervicovaginal microbiota in EC patients (Table 2). Samples were aseptically collected from the vagina or cervix during surgery or preoperatively using swabs or from tissue biopsies. All these studies compared the cervicovaginal microbiota of patients with cancer or EH to that of healthy individuals or patients with benign uterine conditions. EC patients did not exhibit significant alterations in α-diversity compared to controls. Only Semertzidou et al. reported an increase in α-diversity in the EC group [27], while Hakimjavadi et al. reported an increase only in patients with high-grade EC [28].

Across all these studies, β-diversity was significantly increased between cases and controls, indicating a significant shift in microbial composition associated with malignancy. Cervical and vaginal environments in EC were less frequently dominated by *Lactobacillus* spp. and instead colonized by complex, polymicrobial anaerobic communities [27,28,29,30,31,32]. Specifically, Barczyński et al. and Walther-António et al. reported reductions in *Lactobacillus iners* and *L. crispatus*, respectively [30,32], accompanied by increases in *Mobiluncus curtisii*, *Dialister pneumosintes*, *Atopobium vaginae*, and *Porphyromonas* spp. Similarly, Semertzidou et al. demonstrated consistent depletion of *L. crispatus* in EC patients across multiple anatomical sites, coupled with overrepresentation of *Prevotella*, *Peptoniphilus*, and *Porphyromonas* [27].

The presence and abundance of specific bacterial taxa were frequently associated with tumor type, grade, and host factors. Walsh et al. identified *Porphyromonas somerae* in 100% of women with type II EC, suggesting a subtype-specific microbial signature [31]. Hakimjavadi et al. showed that microbial community state types (CSTs) varied significantly across tumor grades and histological subtypes, with Fusobacterium nucleatum and *Prevotella bivia* enriched in high-grade and serous tumors (CST IV) and L. gasseri (CST II) reduced across the EC cohort. These CSTs also correlated with inflammatory gene expression and predicted tumor phenotype with high accuracy (AUC = 0.88) [28].

### 3.3. Functional and In Vitro Studies

While the observational studies we included provided crucial insights into the microbial composition in EC, functional and in vitro studies are essential to understand the mechanistic roles of specific microbial taxa in tumor initiation and progression. This group of studies goes beyond association, exploring how selected microbes interact with host tissues and influence cellular behavior. We included only three relevant papers (Semertzidou et al., Caselli et al. and Crooks et al.).

Semertzidou et al. demonstrated a potential protective role for *Lactobacillus crispatus*, showing that its conditioned media significantly inhibited the proliferation of EC organoids in vitro. This finding suggests that beyond association, specific members of the commensal flora may exert direct anti-tumor effects, potentially via modulation of epithelial immunity, pH, or metabolite secretion [27].

Caselli et al. and Crooks et al. provided mechanistic insights into how *Porphyromonas somerae* may influence the development or progression of endometrial cancer: through direct invasion of epithelial cells, *P. somerae* may evade immune surveillance and modulate host intracellular metabolism, including the production of succinate, a metabolite known to stabilize HIF-1α, a driver of cancer progression. Furthermore, they showed that *P. somerae* induces pro-inflammatory cytokines, potentially creating a tumor-promoting environment within the endometrium [33,34].

## 4. Discussion

Across most of the included studies, the α-diversity of endometrial microbiota was significantly increased in EC; these findings suggest a shift toward a more polymicrobial environment potentially driven by inflammation, immune modulation, or epithelial barrier disruption within the endometrial niche. Some studies reported conflicting results; there could be several reasons underlying these discrepancies. One possible explanation is the inclusion of patients with different characteristics regarding menopause status and BMI. In fact, Walsh et al. demonstrated that these factors significantly influence the diversity of the vaginal microbiota [31]. Another reason could be using different indices to measure α-diversity (such as the Shannon index, Chao index, and Pielou’s evenness). Finally, we believe that tumor grading could also be one of the contributing factors. Hakimjavadi suggests that early lesions might maintain *Lactobacillus* dominance (lower α-diversity), while advanced tumors may exhibit microbial infiltration and immune-mediated breakdown, increasing α-diversity. These reasons also apply to cervicovaginal α-diversity, as no consistent trend in α-diversity was observed across the studies analyzed in relation to endometrial cancer. In contrast to the heterogeneity observed in α-diversity, endometrial and cervicovaginal β-diversity consistently distinguished EC-associated microbial communities from benign controls. Nearly all reviewed studies reported statistically significant differences in microbial community structure based on Bray–Curtis dissimilarity, UniFrac metrics, or Aitchison distances. The observed increase in β-diversity reflects the taxonomic signatures reported in the studies we reviewed: Both the endometrial and cervicovaginal environments consistently show a depletion of protective commensals such as *Lactobacillus crispatus* and *L. iners* and an environment of pro-inflammatory and potentially pathogenic taxa. In most of the studies we reviewed, these pathogenic taxa identified in the female reproductive tract were primarily composed of *Prevotella*, *Atopobium*, *Peptoniphilus*, *Dialister*, and *Porphyromonas*. Between these genera, the most studied is *Porphyromonas*, specifically *P. somerae*, which is capable of invading endometrial cells in vitro and inducing the expression of pro-inflammatory cytokines (IL1α, IL1β, IL17α, TNFα), which are associated with inflammatory processes and possibly with endometrial tumor progression.

These findings are further enriched by the emerging literature on the gut microbiota’s role in EC pathogenesis. Under normal conditions, gut microbial communities maintain a dynamic equilibrium; however, disruptions due to factors like obesity, diabetes, and hormonal imbalance, common in EC patients, lead to dysbiosis characterized by a decline in beneficial microbes and an expansion of opportunistic or pro-inflammatory taxa like Firmicutes and Bacteroides. These bacteria producing β-glucuronidase enzymes can deconjugate estrogens in the intestine, promoting reabsorption and elevating systemic estrogen levels, a known risk factor of EC [6,35]. Together, these results suggest that this shift from beneficial to pro-inflammatory bacteria, whether within the gut or the female reproductive tract, may not be isolated phenomena but rather interconnected reflections of broader systemic dysregulation. Accordingly, clarifying the determinants of microbiome composition across body sites is essential; beyond the factors noted above, three domains are especially influential: (1) Diet, which consistently remodels gut taxa and functions across Mediterranean, high-fiber/plant-based, ketogenic, high-protein, and Western patterns, is associated with corresponding shifts in SCFAs, bile acids, and other metabolites [36]. (2) Host genetic background, host genetic variants are associated with structural and functional traits of the gut microbiota [37,38]. (3) Geographic/environmental context, with harmonized global datasets showing robust, region-specific community signatures that can even predict a sample’s world region of origin [39]. Therefore, these elements must be explicitly considered when designing and interpreting studies.

This systematic review employed a rigorous selection process, improving result reliability and providing updated insights on the relation between female reproductive tract microbiota and endometrial cancer; furthermore, to our knowledge, this is the first review in the literature that studies this relationship in depth, also because most of the studies we included were published in the last four years. However, several limitations must be acknowledged: (1) The study is a systematic review based on observational studies. This type of design does not allow for establishing a causal link between microbiota and endometrial cancer. It is difficult to determine whether microbial alterations are a cause or a consequence of the disease. (2) Heterogeneity of the included studies. Sample collection, storage, and preparation methods varied widely across studies, as did the different α- and β-diversity indices and distance metrics used to quantify community structure. These factors may impact the microbial composition detected. (3) Two studies analyzed only DNA (16S or shotgun). This is the standard practice for compositional studies. This limitation arises when trying to understand real-time metabolic activity: in that case, it would be better to use RNA (metatranscriptomics). DNA, on the other hand, may include traces of non-viable microorganisms, potentially overestimating certain species. (4) The pooling of EC with EH in a subset of primary studies. EH represents a biologic continuum that may differ meaningfully from invasive EC, and EC outcomes may vary by grade and histotype. Incomplete and heterogeneous reporting in several studies precluded consistent stratification; pooled EC + EH findings should therefore be interpreted with caution. (5) We note that opportunistic infections (e.g., bacterial vaginosis or subclinical endometritis) can transiently inflate α-diversity and shift community structure, confounding associations with endometrial cancer. To limit confounding from opportunistic infections and transient perturbations of the genital-tract microbiota, most primary studies screened and excluded women with active genital infections and recent antimicrobial/hormonal exposure, standardized pre-operative sterile swab collection (often before routine douching), and reported aseptic handling; for example, Wang et al. [23]. excluded genital infections, recent antibiotics, and intercourse/douching within 48 h, while Barczyński et al. [32] sampled intraoperatively before the betadine douche and excluded recent intravaginal infections; pre-operative sterile sampling was also used by Walsh et al. [31].

## 5. Conclusions

This systematic review highlights a significant association between alterations in the female genital tract microbiota and endometrial carcinoma. Endometrial cancer appears to be linked to the depletion of *Lactobacillus* spp. and an increased abundance of pro-inflammatory bacterial species. Establishing a causal relationship is of fundamental importance, as modulation of the uterine and vaginal microbiota may represent a promising target for the development of conservative therapeutic strategies in endometrial hyperplasia.

## Figures and Tables

**Figure 1 ijms-26-08877-f001:**
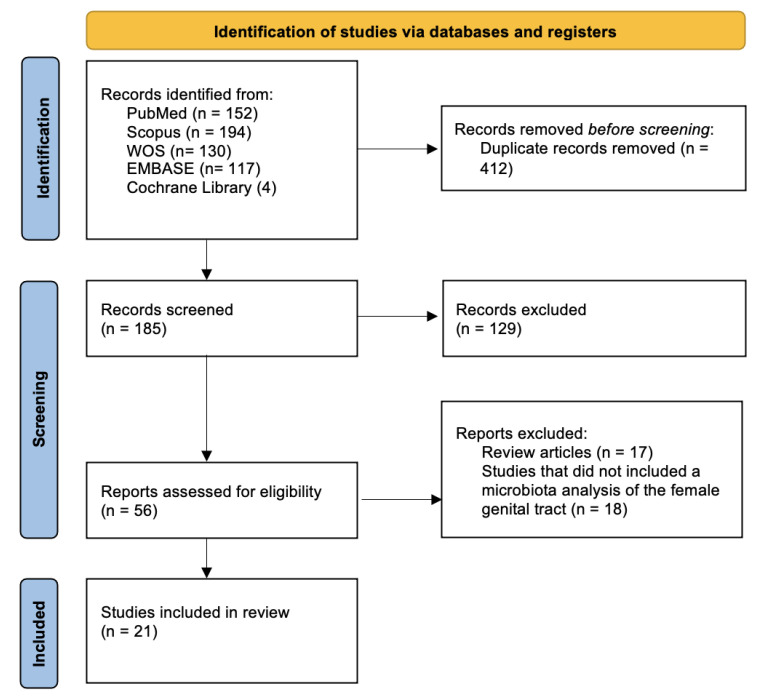
PRISMA flow diagram.

**Table 2 ijms-26-08877-t002:** Methodological characteristics and main results of studies on the cervicovaginal microbiota.

First Author and Year	Sampling Method	Sample Size	Study Results
Gressel et al., 2021 [29]	Sterile swabs from vaginal fornices, ectocervix, rectum, and endometrial cavity during hysterectomy; 16S rRNA.	Controls n = 10, endometrioid EC n = 14, serous EC n = 11.	Significant differences in microbial beta-diversity among niches; USC group showed reduced alpha-diversity in uterine samples and distinct microbial signatures across sites. Cervicovaginal *Lactobacillus* depletion and uterine *Pseudomonas* elevation were biomarkers for USC. Microbiota composition could distinguish USC from controls (*p* = 0.042).
Semertzidou et al., 2024 [27]	Samples collected from the vagina, cervix and endometrium. 16S rRNA sequencing.	EC n = 37 EC, control n = 24	Endometrial cancer patients showed in cervix and rectum reduced *Lactobacillus* (especially *L. crispatus*), increased bacterial diversity, and enrichment of *Porphyromonas*, *Prevotella, Peptoniphilus, Anaerococcus. L. crispatus*-conditioned media had anti-proliferative effect on endometrial organoids.
Walther-António et al., 2016 [30]	Vaginal, cervical, Fallopian, ovarian, and urine samples; 16S rDNA sequencing.	benign n = 10, EH n = 4 (including 1 AEH), EC n = 17.	Cancer and hyperplasia samples showed significantly different microbiome profiles vs. benign. *A. vaginae* and *Porphyromonas* sp. correlated with EC, especially with high vaginal pH. Increased alpha diversity in EC and EH compared to benign. Suggests microbial contribution to tumorigenesis.
Walsh et al., 2019 [31]	Vaginal and cervical swabs obtained preoperatively, 16S rRNA sequencing.	EC n = 66, benign n = 75, HP n = 7.	Postmenopausal status, obesity, and high vaginal pH significantly increased vaginal microbiome diversity. *Porphyromonas somerae* was most enriched in EC patients and proposed as a potential biomarker (AUC = 76.7%). *P. somerae* detected in 100% of Type II EC cases. No differences regarding α-diversity between cancer and control group.
Hakimjavadi et al., 2022 [28]	Vaginal swabs taken pre-surgery using sterile swabs; DNA extracted and shotgun metagenomic sequencing performed.	Benign n = 1, low-grade EC n = 30, high-grade EC n = 20.	Microbial α- and β-diversity correlated with tumor grade. *Fusobacterium ulcerans* and *Prevotella bivia* were enriched in high-grade EC. Vaginal microbiome profiles predicted cancer presence and grade with high accuracy (AUC up to 0.88). No differences regarding α-diversity between cancer and control group.
Barczyński et al., 2023 [32]	Cervical and vaginal swabs collected intraoperatively post-anesthesia, pre-douching; 16S rRNA sequencing.	EC n = 48, hyperplasia n = 21, benign n = 27.	Significant cervicovaginal dysbiosis observed in EC patients. *Mobiluncus curtisii* and *Dialister pneumosintes* were more frequent in cancer patients, while *Lactobacillus iners* was more common in benign cases. Suggests potential role of microbiota in carcinogenesis.

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
