# Peer review of "The Female Reproductive Tract Microbiota and Endometrial Cancer: A Systematic Review"

_ijms, 2025, doi:10.3390/ijms26188877_

Round 1

Reviewer 1 Report

Comments and Suggestions for Authors

Direct recommendation: Minor revision. The review is timely and clinically relevant, but requires brief clarifications (diversity metrics), clearer tables, acknowledgment of EC vs EH pooling, asepsis/contamination controls in low-biomass sampling, and minor heading/English edits to maximize rigor and readability for gynecology/oncology audiences.

  • Introduction: Add one sentence summarizing the pathophysiology linking the gut microbiota to endometrial cancer, highlighting the estrobolome and microbial β‑glucuronidase that can increase enterohepatic estrogen recirculation and potentially raise EC risk, framed as hypothesis‑generating based on emerging evidence..

  • Methods (Section 2.2): Define alpha diversity as within-sample richness/evenness and beta diversity as between-sample dissimilarity, and state how group separation was assessed to enhance interpretability across heterogeneous primary studies.

  • EC and EH pooling: Note explicitly that pooling EC with EH is a limitation; where possible, briefly differentiate EH without atypia vs atypical/EIN, and EC by grade/histotype, or state when data preclude stratification to avoid overgeneralization.

  • Tables 1 and 2: Reformat with clear horizontal rules and dedicated columns so each data point is attributable to its specific study (author/year; site/sample; n per group; method; alpha result; beta result; key taxa; clinical correlation; contamination controls), as the current layout makes attribution difficult.

  • Asepsis/contamination: In Methods, add one sentence confirming attention to aseptic sampling and low-biomass contamination controls in included studies (e.g., sterile, lubricant-free speculum, cervical cleansing, negative controls/blank extractions) and indicate that findings from studies without controls were interpreted cautiously.

  • Heading/English edits: Correct the final heading to “Conclusions,” ensure numbering is sequential, and make minor English edits for clarity.

This is a strong and timely synthesis that advances the field; congratulations on the comprehensive review, and encouragement is offered to pursue more robust, prospectively designed clinical studies to deepen mechanistic understanding and refine the translational pathway for endometrial cancer research.

Author Response

REVIEWER 1, POINT 1

Introduction: Add one sentence summarizing the pathophysiology linking the gut microbiota to endometrial cancer, highlighting the estrobolome and microbial β glucuronidase that can increase enterohepatic estrogen recirculation and potentially raise EC risk, framed as hypothesis generating based on emerging evidence.

Thank you for this general suggestion. As recommended, we have added a sentence to better summarize the hypothesized pathophysiologic link between gut microbiota and endometrial cancer. The revised text in page 2 lines 69-72 now reads: “This concept aligns with the emerging notion of the estrobolome, the collection of gut microbial genes capable of metabolizing estrogens, whereby increased microbial β-glucuronidase activity may enhance enterohepatic recirculation of estrogens and, in turn, hypothetically contribute to endometrial carcinogenesis”

REVIEWER 1, POINT 2

Methods (Section 2.2): Define alpha diversity as within-sample richness/evenness and beta diversity as between-sample dissimilarity, and state how group separation was assessed to enhance interpretability across heterogeneous primary studies.

Thank you for this helpful suggestion. We revised Section 2.2 to standardize terminology and clarify how group separation was assessed across heterogeneous studies. We added the following text on page 3 lines 121-126: “We harmonized diversity terminology across studies by defining α-diversity as with-in-sample richness/evenness (e.g., Shannon, Simpson, Chao1, Pielou’s evenness) and β-diversity as between-sample community dissimilarity (e.g., Bray–Curtis, Jaccard, weighted/unweighted UniFrac, Aitchison). For synthesis, we recorded the index/metric and classified results as ‘higher’, ‘lower’, or ‘no difference’ versus controls.”

REVIEWER 1, POINT 3

EC and EH pooling: Note explicitly that pooling EC with EH is a limitation; where possible, briefly differentiate EH without atypia vs atypical/EIN, and EC by grade/histotype, or state when data preclude stratification to avoid overgeneralization.

We agree that pooling endometrial cancer (EC) with endometrial hyperplasia (EH) may overgeneralize heterogeneous entities. As suggested, we have revised the Discussion section. In the paragraph addressing the study limitations, we have added the following text (page 12 lines 328-332): “The pooling of EC with EH in a subset of primary studies. EH represents a biologic continuum that may differ meaningfully from invasive EC, and EC outcomes may vary by grade and histotype. Incomplete and heterogeneous reporting in several studies precluded consistent stratification; pooled EC+EH findings should therefore be interpreted with caution.”

We have now differentiated EH without atypia versus atypical hyperplasia wherever the primary studies reported it, and updated the summary table to include a dedicated entry for the number of EH with atypia when specified. For EC, only one included study reported tumor grade [28], also histotype was reported by only one study [29]; therefore, stratified synthesis by grade/histotype was not feasible.

REVIEWER 1, POINT 4

Tables 1 and 2: Reformat with clear horizontal rules and dedicated columns so each data point is attributable to its specific study (author/year; site/sample; n per group; method; alpha result; beta result; key taxa; clinical correlation; contamination controls), as the current layout makes attribution difficult.

Thank you for the helpful suggestion. We agree with the rationale; however, a fully expanded layout with dedicated columns for site/sample, n per group, method, alpha result, beta result, key taxa, clinical correlation, and contamination controls would exceed the page width permitted for the main text and markedly reduce legibility.

To balance clarity with space constraints, we have:

  • reformatted Tables 1–2 with clear horizontal rules and one study per row;
  • added a dedicated “Sample size (per group)” column;
  • retained a single “Results (summary)” column.

Because outcomes, indices, and taxa are highly heterogeneous across studies, splitting these into multiple separate columns would require numerous additional columns and still leave some cells empty or ambiguous. If the Editors prefer a fully disaggregated format, we can provide an expanded version with all requested dedicated columns as Supplementary Material.

REVIEWER 1, POINT 5

Asepsis/contamination: In Methods, add one sentence confirming attention to aseptic sampling and low-biomass contamination controls in included studies (e.g., sterile, lubricant-free speculum, cervical cleansing, negative controls/blank extractions) and indicate that findings from studies without controls were interpreted cautiously.

We appreciate the Reviewer’s suggestion. We have revised the Methods to explicitly state our attention to aseptic sampling and low-biomass contamination controls across included studies. All studies reported aseptic sampling; most also reported explicit low-biomass contamination controls, while only seven studies did not reported low-biomass contamination workflow.

We added the following sentence in Methods (page 3 lines 134-138): “To minimize bias inherent to low-biomass microbiome studies, we prioritized reports using aseptic sampling (e.g., sterile, lubricant-free speculum with cervical cleansing) and laboratory contamination controls (negative controls/blank extractions), and we interpreted findings from studies that did not report these measures cautiously in the synthesis.”

REVIEWER 1, POINT 6

Heading/English edits: Correct the final heading to “Conclusions,” ensure numbering is sequential, and make minor English edits for clarity.

Thank you for the helpful suggestion. We have corrected the final section heading to “Conclusions”, verified and fixed sequential numbering of all section and subsection headings (and updated cross-references to sections, figures, and tables accordingly), and performed minor English edits to improve clarity and consistency.

REVIEWER 1, POINT 7

The English could be improved to more clearly express the research.

We thank the reviewer for this observation. The manuscript has undergone a thorough language revision to improve clarity, grammar, and readability, ensuring that the research is expressed more clearly and effectively.

Reviewer 2 Report

Comments and Suggestions for Authors

The authors have presented a well-written and systematic review. This review may be helpful in encompassing rapidly growing findings associated with altered genital tract microbiota and endometrial cancer, which could have important implications for clinical practice, diagnosis, understanding disease mechanisms, and guiding future research.

However, I have several suggestions for strengthening the manuscript.

  1. The authors are encouraged to include a discussion of the disadvantages or limitations related to the use of microbiota in clinical or research contexts; it would be valuable to specifically discuss the role of opportunistic infections and their impact on microbiota diversity.
  2. The inclusion of summary tables that briefly display the parameters used to identify the microbiota including bacteria, viruses, and fungi. This would greatly improve clarity and facilitate comparison across studies.
  3. The review would benefit from a discussion of the influence of diet, host genetics, and geographical distribution on the human microbiota, as these are key factors that shape community composition and diversity in both health and disease.

Addressing these points would enhance the rigor, relevance, and utility of the review for both research and clinical audiences.

Author Response

REVIEWER 2, POINT 1

The authors are encouraged to include a discussion of the disadvantages or limitations related to the use of microbiota in clinical or research contexts; it would be valuable to specifically discuss the role of opportunistic infections and their impact on microbiota diversity.

We thank the reviewer for this valuable suggestion. We have added the following text to the Discussion section (page 12, lines 332–342): “We note that opportunistic infections (e.g., bacterial vaginosis or subclinical endometritis) can transiently inflate α-diversity and shift community structure, confounding associations with endometrial cancer. To limit confounding from opportunistic infections and transient perturbations of the genital-tract microbiota, most primary studies screened and excluded women with active genital infections and recent antimicrobial/hormonal exposure, standardized pre-operative sterile swab collection (often before routine douching), and reported aseptic handling; for example, Wang et al. excluded genital infections, recent antibiotics, and intercourse/douching within 48 h, while BarczyÅ„ski et al. sampled intraoperatively before the betadine douche and excluded recent intravaginal infections; pre-operative sterile sampling was also used by Walsh et al.”

REVIEWER 2, POINT 2

The inclusion of summary tables that briefly display the parameters used to identify the microbiota including bacteria, viruses, and fungi. This would greatly improve clarity and facilitate comparison across studies.

We thank the reviewer for this important suggestion. As recommended, we have created a comprehensive summary table outlining the parameters used to identify the microbiota (including bacteria, viruses, and fungi) across the included studies. This table has been added to the Supplementary Materials to improve clarity and facilitate direct comparison among studies.

REVIEWER 2, POINT 3

The review would benefit from a discussion of the influence of diet, host genetics, and geographical distribution on the human microbiota, as these are key factors that shape community composition and diversity in both health and disease.

We thank the reviewer for this insightful suggestion. We agree that diet, host genetics, and geographical distribution are key determinants of microbiota composition and diversity. To address this, we added a concise paragraph to the Discussion. Given the already substantial length of the Discussion, we deliberately kept this addition brief to preserve readability. However, if the reviewer consider it necessary, we would be pleased to expand this subsection with a more detailed appraisal and additional references. We added the following text in page 12 lines 303-312: “Accordingly, clarifying the determinants of microbiome composition across body sites is essential; beyond the factors noted above, three domains are especially influential: (1) Diet, which consistently remodels gut taxa and functions across Mediterranean, high-fibre/plant-based, ketogenic, high-protein, and Western patterns, is associated with corresponding shifts in SCFAs, bile acids, and other metabolites [37]. (2) Host genetic background, host genetic variants are associated with structural and functional traits of the gut microbiota [38,39]. (3) Geographic/environmental context, with harmonized global datasets showing robust, region-specific community signatures that can even predict a sample’s world region of origin [40]. Therefore, these elements must be explicitly considered when designing and interpreting studies.”

Reviewer 3 Report

Comments and Suggestions for Authors

The manuscript deals with the relationship between the reproductive tract microbiota and endometrial cancer, which is indeed an important and fast-developing topic. However, in its current form the review feels rather limited and not as informative as it could be. The number of studies included and cited is relatively small, the coverage of mechanisms is brief, and key areas such as fungal and viral communities, methodological challenges, and confounding clinical factors are underdeveloped. As a result, the review does not yet provide readers with a comprehensive understanding of the field.

1-Only around 40 references are included, which is too few for a systematic review. This gives the impression of a narrow literature search.

2-The manuscript structure needs editing (two “Discussion” sections, inconsistent terminology).

3-Figures and tables could be much more informative — for example, a summary figure mapping taxa to tumor features, or a methodological quality matrix.

Author Response

REVIEWER 3, POINT 1

Only around 40 references are included, which is too few for a systematic review. This gives the impression of a narrow literature search.

We appreciate this thoughtful comment. The relatively small number of references reflects the nascency of the uterine/ endometrial microbiome field, particularly for low-biomass sites such as the endometrium, where robust sequencing and contamination-aware workflows have only recently matured. Foundational signals linking the uterine microbiome to endometrial cancer (EC) appeared only in 2016 and 2019, with subsequent growth of primary studies mainly from 2021 onward; as a result, most eligible EC/EH papers are recent, and the pool remains limited despite a comprehensive search strategy.

Importantly, several of these newer studies also illustrate why the literature base is still compact: they underscore the technical and methodological challenges of low-biomass sampling and the need for stringent controls, which have only recently been standardized across centers [Leoni et al., 2024: 10.3390/microorganisms12061090].

To address your concern and better contextualize our findings, we added four references on the Discussion section.

REVIEWER 3, POINT 2

The manuscript structure needs editing (two “Discussion” sections, inconsistent terminology).

Thank you for the important suggestion. We have revised the final section heading to “Conclusions,” ensured that all section and subsection headings follow the correct sequential numbering (with corresponding updates to cross-references for sections, figures, and tables), and made minor English edits to enhance clarity and consistency.

REVIEWER 3, POINT 3

Figures and tables could be much more informative — for example, a summary figure mapping taxa to tumor features, or a methodological quality matrix.

Thank you for this helpful suggestion. Since all the included studies are cross-sectional in design, we applied the JBI Analytical Cross-Sectional appraisal matrix to assess their methodological quality, we have provided the full evaluation in the Supplemental Materials. In addition, we revised Tables 1 and 2 to improve clarity and readability, ensuring that the data are more clearly attributable to each study.

Round 2

Reviewer 2 Report

Comments and Suggestions for Authors

Authors,

I have no further comments. Your explanations have adequately resolved my queries and are satisfactory.

With best

Abhishek Kumar

Reviewer 3 Report

Comments and Suggestions for Authors

I am satisfied with the author's revised manuscript.